# A Prospective Study: Highlights of Hippocampal Spectroscopy in Cognitive Impairment in Patients with Type 1 and Type 2 Diabetes

**DOI:** 10.3390/jpm11020148

**Published:** 2021-02-19

**Authors:** Julia Samoilova, Mariia Matveeva, Olga Tonkih, Dmitry Kudlau, Oxana Oleynik, Aleksandr Kanev

**Affiliations:** 1Medical Faculty, Siberian State Medical University, 634050 Tomsk, Russia; samoilova_y@inbox.ru (J.S.); ostonkih@mail.ru (O.T.); oleynikoa@mail.ru (O.O.); alexkanev92@gmail.com (A.K.); 2Institute of Immunology, Federal Medical and Biological Agency of Russia, 115478 Moscow, Russia; dakudlay@generiumzao.ru

**Keywords:** type 1 diabetes mellitus, type 2 diabetes mellitus, proton spectroscopy, hippocampus, cognitive dysfunction

## Abstract

Diabetes mellitus type 1 and 2 is associated with cognitive impairment. Previous studies have reported a relationship between changes in cerebral metabolite levels and the variability of glycemia. However, the specific risk factors that affect the metabolic changes associated with type 1 and type 2 diabetes in cognitive dysfunction remain uncertain. The aim of the study was to evaluate the specificity of hippocampal spectroscopy in type 1 and type 2 diabetes and cognitive dysfunction. Materials and methods: 65 patients with type 1 diabetes with cognitive deficits and 20 patients without, 75 patients with type 2 diabetes with cognitive deficits and 20 patients without have participated in the study. The general clinical analysis and evaluation of risk factors of cognitive impairment were carried out. Neuropsychological testing included the Montreal Scale of Cognitive Dysfunction Assessment (MoCA test). Magnetic resonance spectroscopy (MRS) was performed in the hippocampal area, with the assessment of *N*-acetylaspartate (NAA), choline (Cho), creatine (Cr), and phosphocreatine (PCr) levels. Statistical processing was performed using the commercially available IBM SPSS software. Results: Changes in the content of NAA, choline Cho, phosphocreatine Cr2 and their ratios were observed in type 1 diabetes. More pronounced changes in hippocampal metabolism were observed in type 2 diabetes for all of the studied metabolites. Primary risk factors of neurometabolic changes in patients with type 1 diabetes were episodes of severe hypoglycemia in the history of the disease, diabetic ketoacidosis (DKA), chronic hyperglycemia, and increased body mass index (BMI). In type 2 diabetes, arterial hypertension (AH), BMI, and patient’s age are of greater importance, while the level of glycated hemoglobin (HbA1c), duration of the disease, level of education and insulin therapy are of lesser importance. Conclusion: Patients with diabetes have altered hippocampal metabolism, which may serve as an early predictive marker. The main modifiable factors have been identified, correction of which may slow down the progression of cognitive dysfunction.

## 1. Introduction

Clinical studies indicate that type 1 and type 2 diabetes are risk factors for cognitive decline, while structural and functional deficits are associated with synaptic plasticity processes. Experimental data derived from rodent model of diabetes demonstrate the decline in learning and memory of varying extent, owing to the hippocampus dysfunction. These changes are associated with an increase in oxidative stress, levels of pro-inflammatory cytokines, β-amyloid, as well as dysfunction of the hypothalamic-pituitary-adrenal axis [1]. The hippocampus is a vital structure for learning and memory. High density of insulin receptors has been found in this area of the brain [2]. Magnetic resonance spectroscopy (MRS), the latest biochemical analysis method, detects changes in metabolic neurochemical levels and energy metabolism in different areas of the brain in vivo [3,4,5]. Considering that MRS is both safe and non-invasive, researchers have used MRS to investigate the underlying causes of many neurological conditions, such as Alzheimer’s disease and diabetes mellitus [6]. Monitoring changes in the level of each metabolite and metabolite ratios can provide information on neuronal damage, membrane metabolic dysfunctions, and transmission defects that occur in neurological diseases [7,8]. The purpose of the present study was to evaluate the features of spectroscopy of the hippocampus in patients with type 1 and 2 diabetes. Hypothesis: patients with type 1 and type 2 diabetes have risk factors for impaired neurometabolism in the hippocampus, which is clinically manifested by cognitive impairment.

## 2. Materials and Methods

All patients included in the study signed an informed consent, which was approved by an ethics committee (No. 265 from 2 May 2017). Sixty-five patients with type 1 diabetes with cognitive deficits and 20 patients without, 75 patients with type 2 diabetes with cognitive deficits and 20 patients without have participated in the study. According to the design, the study is characterized as an observational, cross-sectional, single-center, continuous, comparative study. Inclusion criteria for the study: diabetes type 1 and 2, age 18–65 years with varying degrees of disease compensation, carbohydrate metabolism, severity of vascular changes, the duration of the disease, and the type of therapy; obligatory presence of voluntary informed consent. Exclusion criteria: non-compliance with the inclusion criteria; presence of organic brain damage (tumors, stroke); of drugs or substances that alter cognitive functions (psychotropic, narcotic drugs); chronic alcoholism (anamnestic, ambulatory history); vitamin B12 deficiency (determined at study inclusion); condition after severe trauma and surgery; presence of hematologic, oncologic, and severe infectious diseases; decompensation of chronic heart failure with pronounced clinical symptoms, functional class of chronic heart failure functional class higher than II; acute coronary syndrome and transient ischemic attack in previous 6 months.

### 2.1. Sample Characteristics

The cohort of patients examined is presented below (Table 1).

Patients with type 1 diabetes were younger and had longer duration of the disease. However, inside of type 1 and type 2 diabetes groups, no statistically significant differences in patients’ characteristics were noted between subgroups with and without cognitive dysfunction.

### 2.2. Risk Factors

Each patient underwent the assessment of risk factors for cognitive impairment, including metabolic parameters, acute complications of diabetes, hypertension, Body Mass Index (BMI), duration of the disease, age, level of education, smoking status, alcohol abuse, insulin therapy.

### 2.3. Cognitive Function

For the diagnosis of cognitive impairment, the generally accepted test is the Montreal cognitive assessment (MoCA) (Copyright 2019, Ziad Nasreddine, MD) [9]. The test assesses eight cognitive domains: executive and visual-constructive skills, naming, memory, attention, speech, abstraction, delayed memory, and orientation. The maximum score is 30 points; borderline—26 points. The specificity of MoCA in detecting mild cognitive impairment is 90%, the sensitivity is 87%. A MoCA survey takes up to 10 min. This scale is now recommended by most modern experts in the field of cognitive impairment for widespread use in everyday clinical practice.

### 2.4. Proton Spectroscopy of the Brain

MRS of the brain with an echo time (TE) of 135 ms, and the volume of one voxel of 1.5 cm^3^ was performed immediately after MRI of the brain. This technique was carried out in a multi-voxel mode, which allows placing 64 voxels on one slice simultaneously. In the areas of interest, the main spectra of *N*-acetylaspartate (NAA), choline (Cho), creatine (Cr), and phosphocreatine (PCr), as well as their ratios, were recorded. The protocol of proton magnetic resonance spectroscopy of the brain included the following steps (the total time of the study is approximately 35–40 min): (1) positioning of the examined patient (supine); (2) conducting a standard MRI of the brain; (3) performing sighting slices on the hippocampus area, T2 3 mm hippocampus; (4) MRS on the area of interest—frontal and temporal lobes area of the hippocampus (the camera serial interface protocol was used); (5) obtaining MR spectra of NAA, Cho, Cr, PCr in multivoxel mode; (6) postprocessing of the results of proton magnetic resonance spectroscopy including the analysis of spectrograms and the construction of color maps of the distribution of the main metabolites, as well as their ratios. The Turbo Spin Echo (Turbo SE) method with the parameters: recovery time—1500 ms, TE—135 ms, field of view—160 mm, matrix—192 × 256, slice thickness—5 mm, scanning time—12 min, was used. The quantitative characteristics of the studied metabolites and their ratios in the gray matter of the cerebral cortex, white matter of the brain, subcortical structures, and in the hippocampus on both sides were assessed. Using a regional approach, data were selected for NAA, Cho, Cr, PCr, localized in the left and right hippocampus, as shown in Figure 1. There were no missing data in our selection of patients.

To calculate sample size, we used the formula calculating the minimum sample size. As previous studies investigating hippocampal metabolism parameters in diabetes patients are lacking, the required sample size was estimated based on the results of two studies that were assessing the NAA/Cr ratio in patients with type 2 diabetes mellitus [10] (NAA/Cr ratio 1.36 ± 0.15) and mild cognitive impairment, which also included diabetes patients (NAA/Cr ratio 1.58 ± 0.14) [6]. According to the aforementioned data, should we pursue the power of 0.9, the sample size of 11 in each group would be sufficient to prove the initial hypothesis.

The statistical processing was performed using the IBM SPSS Statistics software (USA), 19.0.0 Russian version. The following coefficients were evaluated: Shapiro–Wilk *W*-test, distribution estimate. Differences were determined by Student’s *t*-test for normal, Mann–Whitney *Z*-test for non-normal distribution, Wilcoxon’s test for dependent data. Descriptive analysis included the determination of the arithmetic mean X, error of the mean m, and calculation of median and quartiles (Me, Q1–Q3) depending on the type of distribution. The critical level of significance *p* when testing statistical hypotheses in the study was taken equal to 0.05. Qualitative data were analyzed using frequency analysis. For definition of accuracy, we used Pearson *χ^2^* test. Spearman test was used for the correlation analysis [11].

## 3. Results

As a result of randomization, 15 people each from the group with type 1 and type 2 diabetes dropped out of the study (Figure 2).

In patients with type 1 diabetes presenting with cognitive impairment, an increase in NAA and PCr, and a decrease in the Cho content was noted. NAA/Cho, Cho/Cr ratios were also decreased, while the Cho/Cr ratio was increased. In patients with type 2 diabetes, more pronounced metabolic impairments were evidenced (Table 2).

The correlation analysis was carried out between the content of basic metabolites in brain cells and the basic indicators of glycemic variability (Figure 3).

As a result of the study on the relationship between metabolite content and glycemic variability indices, a number of significant positive correlations were registered: the level of NAA with the index of hypoglycemic risk/low blood glucose index (LBGI), Cho with the index of glycemic lability/liability index (LI), the duration and the time in range level (TIR), Cr with the index of prolonged glycemic increase (CONGA), and PCr with LI, CONGA and TIR.

In addition, the elevated HbA1c level, the glycemic lability index, and the mean glucose value were correlated in patients with Type 2 diabetes with reduced creatine levels in the hippocampus region. A decrease in the Cho creatine to NAA/Cr ratio was recorded in patients with increased LI and average glycemia level, when the Cho/Cr increase was more frequently noted when the long-term glycemic increase index (CONGA) was increased (Table 3).

When assessing the influence of risk factors for cognitive impairment on brain metabolism in type 1 diabetes, there was a relationship between NAA left and HbA1c level (−0.247, *p* ≤ 0.05), Сho on the right and age, history of diabetic ketoacidosis and degree of cognitive impairment (−0.237, −0.230, 0.216, *p* ≤ 0.05), Cr on the right and a history of severe hypoglycemia (0.220, *p* ≤ 0.05), Cr2 on the left and body mass index (−0.218, *p* ≤ 0.05), NAA/Cr on the right and a history of severe hypoglycemia episodes (0.210, *p* ≤ 0.05).

When assessing the influence of risk factors for cognitive impairment on brain metabolism in type 2 diabetes, associations between NAA on the left and HbA1c level (−0.733, *p* ≤ 0.05), NAA on the left and the level of arterial hypertension (0.511, *p* ≤ 0.05), Сho on the left and the level of arterial hypertension (0.682, *p* ≤ 0.05), Сho on the right and age and the degree of cognitive impairment (0.785, 0.576, −0.561, *p* ≤ 0.05), Cr on the right and the duration of the disease and the degree of cognitive impairment (0.445, −0.508, 0.619, *p* ≤ 0.05), Cr2 on the left and the degree of arterial hypertension (0.577694, *p* ≤ 0.05), NAA/Cr on the left and a history of episodes of diabetic ketoacidosis in the anamnesis, and the level of HbA1c (0.451, 0.733, *p* ≤ 0.05), NAA/Cr on the left and the presence of higher education and insulin therapy (0.424, −0.596, *p* ≤ 0.05), NAA/Chо on the left and body mass index, and degree of arterial hypertension (−0.562, −0.481, *p* ≤ 0.05), NAA/Chо on the left and body mass index (−0.529, *p* ≤ 0.05), Chо/Cr on the left and age, and glycemic level (−0.457, −0.733, *p* ≤ 0.05), Chо/Cr on the right and the degree of arterial hypertension (−0.512, *p* ≤ 0.05) were noted.

For the purpose of timely detection of cognitive dysfunction at the earliest stage of the pathological process development, a method of early prediction of cognitive dysfunction in patients with type 1 and 2 diabetes using neuroimaging methods was developed.

For this aim, the data obtained by MRS were processed by the program on the platform of IBM Watson Studio 1.2.2.0, 2018, Armonk, NY, USA, 2018, to build a neural network model, which is a decision support system. This program is used to generalize a large number of complexly structured data using non-linear mathematical operations and allows you to build a model for predicting cognitive disorders in diabetes patients.

The predicted value of the cognitive test was used as the output parameter. The accuracy of this model for type 1 diabetes was 73%, error 1.7%, and for type 2 diabetes—79%, error 1.3%.

## 4. Discussion and Conclusions

With a change in the modern lifestyle, the incidence of diabetes increases from year to year [12]. The prevalence of cognitive dysfunction caused by diabetes is 10.8–65.0%, and its occurrence is associated with hippocampal atrophy [13,14,15]. In patients with diabetes, the literature suggests lower values of NAA of the hippocampus on both sides in patients with diabetic retinopathy, which is in accordance with our results, as in our study, patients with cognitive impairment also had a decrease in this metabolite [16]. These data indicate a common effect of complications of diabetes on metabolism in hippocampal cells, which is associated with the loss of neurons or axons and is independently associated with the development of diabetes.

Obesity combined with type 2 diabetes is an important factor of brain damage. The available knowledge demonstrates that in patients with obesity, there are a number of cerebral disorders, either associated with or preceding diabetes, including impaired substrate processing, insulin resistance, and impaired organ interconnections [17]. In keeping with that, in our study, it is BMI that is a risk factor for neurometabolic disorders.

In experimental models of diabetes, the role of dysglycemia on the change in the level of metabolites of the hippocampus was shown, along with the violation of glucose oxidation and an increase in creatinine levels [18]. Another study demonstrated the acute effect of hyperglycemia, i.e., in fact, DKA, on a decrease in NAA, which was combined with hyperphosphorylation of the tau protein, which supports the results of our study and the role of acute complications in the dysfunction of the hippocampus at the stage of manifestation [19]. Currently, data are scant on the effects of hypoglycemia on hippocampal metabolism. In a recent study, Wiegers et al. performed a two-stage hyperinsulinemic euglycemic (5 mmol/L)—hypoglycemic (2.8 mmol/L) test in 7 patients with poor hypoglycemic awareness and in seven patients with normal awareness of hypoglycemia, Hypoglycemia, and in healthy controls. The results showed that all metabolites were reduced by 20% in patients with poor awareness (*p* < 0.001) [20]. In the same study, patients with DM type 1 and frequent hypoglycemia had altered metabolism of the hippocampus.

Cao et al. investigated brain metabolites in the frontal and parietal cortex in 33 patients with diabetes and hypertension and noted the significant decrease in NAA/Cr and Cho/Cr ratios [21]. In our study, hypertension was an important determinant of changes in the hippocampal region.

Other risk factors, such as duration of the disease and age, naturally impair cognitive function and cannot be corrected [22].

A limitation of the study was the non-randomized nature of their comparisons. However, in the future, these points will be taken into account.

In this study, when performing proton spectroscopy, patients with type 1 diabetes showed an increase of NAA, Cho, and Cr2 levels in the hippocampus area, which are responsible for normal neuronal functioning. In type 2 diabetes, NAA levels were increased, as well, Cho, Cr, and Cr2 content decreased. According to several authors, such changes occur in gliosis and membrane necrosis and when oxidative stress is activated [23,24]. MRS of the brain in patients with diabetes shows changes associated with neurodegenerative processes [25]. In this connection, we determined non-invasive markers of metabolic and structural brain changes in patients with diabetes by means of various MRI techniques.

## Figures and Tables

**Figure 1 jpm-11-00148-f001:**
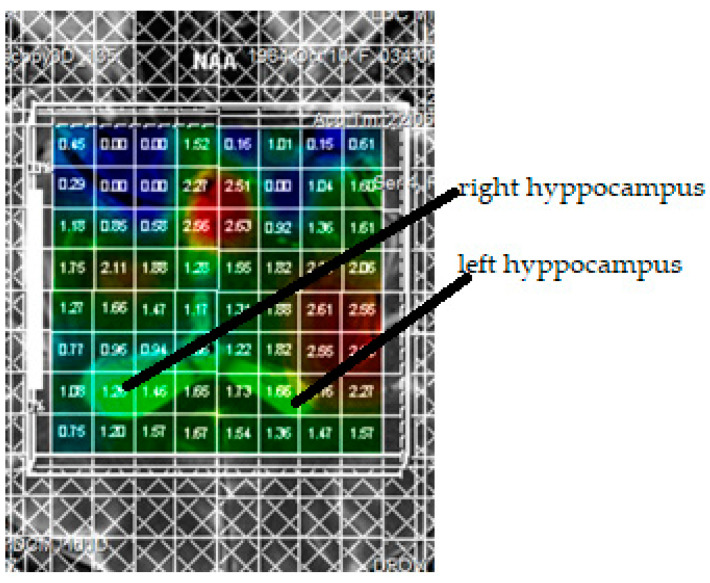
A Regional approach to assessing metabolite content in the hippocampus.

**Figure 2 jpm-11-00148-f002:**
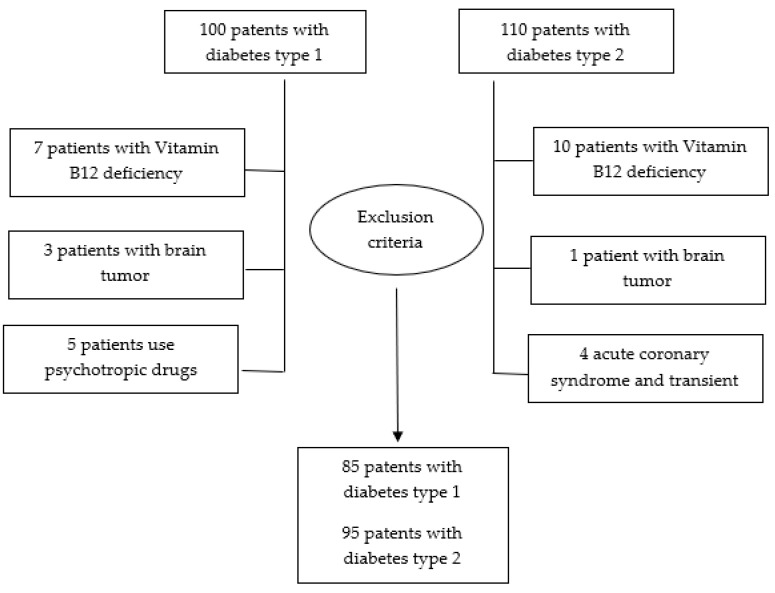
A Flowchart of randomized process.

**Figure 3 jpm-11-00148-f003:**
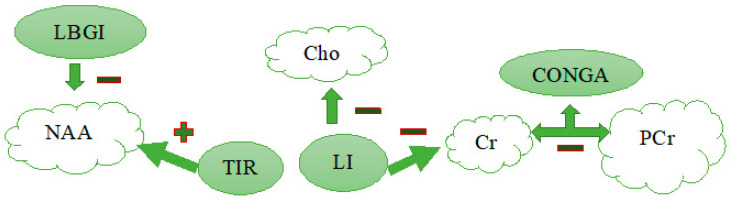
Correlation analysis of the relationship between brain metabolites and glycemic variability indicators. Note. NAA—*N*-acetylaspartate, Cho—choline, Cr—creatine, PCr—creatine phosphate, LBGI—low blood glucose index, LI—lability index, TIR—time in range, CONGA—glycemia long-term increase index, <<+>>—positive correlation, <<−>>—negative correlation.

**Table 1 jpm-11-00148-t001:** Characteristics of patients with type 1 and type 2 diabetes; M (interquartile range—Q1–Q3).

Parameters	Type 1 Diabetes and Cognitive Dysfunction	Type 1 Diabetes and Normal Cognitive Functions	Type 2 Diabetes and Cognitive Dysfunction	Type 2 Diabetes and Normal Cognitive Function
Age, years	44 (42–48)	45 (43–48)	53 (37–69)	53 (31–71)
Disease duration, years	13 (10–21)	13 (6–23)	10 (4–14)	9 (2–13)
HbA1с, %	8.4 (6.6–9.3)	7.4 (5.4–10.1)	8.2 (6.3–9.9)	7.9 (6.2–9.1)
Glycemia, mmol/l	8 (5.6–18.3)	8 (7–9)	7.5 (5.6–9.0)	7.2 (7.0–7.5)

**Table 2 jpm-11-00148-t002:** The content of metabolites of the hippocampus in patients with type 1 and type 2 diabetes with different cognitive functions.

Hippocampal Metabolites	Type 1 Diabetes and Cognitive Dysfunction	Type 1 Diabetes and Normal Cognitive Functions	Type 2 Diabetes and Cognitive Dysfunction	Type 2 Diabetes and Normal Cognitive Function
NAA left	1.796 ± 0.418 *	1.5705 ± 0.317	1.723 ± 0.427 *	0.768 ± 0.472
NAA right	1.851 ± 0.320	1.848 ± 0.204	1.966 ± 0.580 *	0.574 ± 0.158
Cho left	0.854 ± 0.255	0.946 ± 0.088	2.028 ± 1.333 *	0.949 ± 0.223
Cho right	0.919 ± 0.239 *	1.312 ± 0.496	1.119 ± 0.699	1.101 ± 0.257
Cr left	0.901 ± 0.212	0.905 ± 0.190	0.924 ± 0.137 *	0.741 ± 0.233
Cr right	0.900 ± 0.116	0.939 ± 0.136	1.567 ± 0.429 *	0.582 ± 0.246
PCr left	1.419 ± 0.297 *	1.219 ± 0.271	1.162 ± 0.483 *	0.667 ± 0.175
PCr right	2.025 ± 0.723	1.466 ± 0.450	2.147 ± 0.740 *	0.523 ± 0.167
NAA/Cr left	0.544 ± 0.293	0.596 ± 0.161	1.787 ± 0.406 *	0.616 ± 0.226
NAA/Cr right	0.498 ± 0.102	0.513 ± 0.087	1.980 ± 0.913 *	0.536 ± 0.140
NAA/Cho left	0.568 ± 0.217	0.624 ± 0.117	0.970 ± 0.138	0.913 ± 0.388
NAA/Cho right	0.592 ± 0.123 *	0.723 ± 0.304	1.332 ± 0.684 *	1.106 ± 0.342
Cho/Cr left	1.118 ± 0.358 *	1.087 ± 0.249	1.820 ± 1.246	1.700 ± 0.456
Cho/Cr right	1.204 ± 0.216 *	1.437 ± 0.615	1.017 ± 0.184 *	1.998 ± 0.785

Note: * *p* ≤ 0.05.

**Table 3 jpm-11-00148-t003:** Correlation of brain metabolites, carbohydrate metabolism parameters, and serum protein taupe in patients with Table 2. diabetes mellitus.

Parameters	Spearman’s Criterion	*p*
Cr and HbA1с	−0.8	0.007
Cr and LI	−0.7	0.03
Cr and mean	−0.8	0.007
NAA-Cr & mean	−0.6	0.048
Cho-Cr and CONGA	−0.6	0.04
Cho-Cr & LI	−0.6	0.03

Note: NAA—*N*-acetylaspartate, Cho—choline, Cr—creatine, PCr—creatine phosphate, mean—average value of glycemia, LI—glycemia lability index, CONGA—glycemia long-term increase index.

## Data Availability

The data presented in this study are available on request from the corresponding author. The data are not publicly available due to ethical restrictions.

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
