# Peer review of "A Prospective Study: Highlights of Hippocampal Spectroscopy in Cognitive Impairment in Patients with Type 1 and Type 2 Diabetes"

_jpm, 2021, doi:10.3390/jpm11020148_

Round 1

Reviewer 1 Report

Summary of paper

The Authors compared cognitive function by means of hippocampal spectrometry in 180 patients with type 1 (N=85) and type 2 (N=95) diabetes. Comparing patients with cognitive imparment (N=65 with type 1 and N=75 with type 2 diabetes) vs those with normal cognitive function (N=20 with type 1 and N=20 with type 2 diabetes) they found differences in some hippocampal metabolites. This may represent an interesting topic, but the current manuscript has significant issues to be addressed.

Title

- Identify the nature of the study: retrospective?

Methods: study design

- The nature of the study is unclear (retrospective?). More generally the criteria used for patients’ selection remain unexplained.  

- The Authors didn’t take in due account the relevant issue of groups’ comparability. Of concern they failed to provide a table showing the main demographic and clinical characteristics of the four groups of enrolled patients: subjects with and without cognitive impairment and type 1 and type 2 diabetes (see also comment below).

Methods: statistical analysis

- More details on sample size determination should be provided.

- All statistical analyses are unadjusted. This aspect may represent an important limitation, also considering the non-randomized nature of the comparisons.

- How missing data were handled

Results

  • A study flow chart showing the different steps of patients inclusions and exclusions should be present.
  • It is mandatory to add a new table showing the demographic and clinical characteristics of the four patients’ groups considered.

Discussion

- Among the study’s Limitations the Authors should mention the non-randomized nature of their comparisons, and – more generally - the issue of comparability of the patients’ groups, particularly for unmeasured variables. Moreover the Authors should not interpret the results which are statistically significant differences as being clinically important.  

Other comments

- At page 3, line 90  ‘abnormal distribution’ should be ‘non-normal distribution’.

Summary comments

- This study may be of potential interest: however the current version has significant issues to be addressed, needing a profound revision.

Author Response

Point 1:

Title

- Identify the nature of the study: retrospective?

 Response 1: 

This was a prospective study; we followed your recommendation and stated so in the title.

Point 2:

Methods: study design

- The nature of the study is unclear (retrospective?). More generally the criteria used for patients’ selection remain unexplained.  

- The Authors didn’t take in due account the relevant issue of groups’ comparability. Of concern they failed to provide a table showing the main demographic and clinical characteristics of the four groups of enrolled patients: subjects with and without cognitive impairment and type 1 and type 2 diabetes (see also comment below).

 Response 2: 

We inserted a sentence on the study design and the inclusion and exclusion criteria (line 55 67).:

According to the design, the study is characterized as an observational, cross-sectional, single-center, continuous, comparative study. Inclusion criteria for the study: diabetes type 1 and 2, age 18-65 years with varying degrees of of disease compensation, carbohydrate metabolism, severity of vascular changes, the duration of the disease, and the type of therapy; obligatory presence of voluntary informed consent. Exclusion criteria: non-compliance with the inclusion criteria; presence of organic brain damage (tumors, stroke);of drugs or substances that alter cognitive functions (psychotropic, narcotic drugs); chronic alcoholism (anamnestic, ambulatory history); vitamin B12 deficiency (determined at study inclusion); condition after severe trauma and surgery; presence of hematologic, oncologic, and severe infectious diseases; decompensation of chronic heart failure with pronounced clinical symptoms, functional class of chronic heart failure functional class higher than II; acute coronary syndrome and transient ischemic attack in previous 6 months.

-  We have included a table (characteristic of the group - Table 1) -line 68

Point 3:

Methods: statistical analysis

- More details on sample size determination should be provided.

- All statistical analyses are unadjusted. This aspect may represent an important limitation, also considering the non-randomized nature of the comparisons.

- How missing data were handled

Response 3:  No sample size determination analysis was performed, as data was not normally distributed. Tests used to estimate sample size rely on the known distribution pattern. Nonparametric tests are distribution-independent.

It is true that statistical analyses were unadjusted. However, the absence of statistically significant differences in clinical characteristics of patients in the compared groups gives hope that no confounding factors were responsible for the observed results.

Data were thoroughly collected. No cases of missing data were present.

Point 4:

Results

  • A study flow chart showing the different steps of patients inclusions and exclusions should be present.
  • It is mandatory to add a new table showing the demographic and clinical characteristics of the four patients’ groups considered.

Response 4: 

-  We have included flow chart (line 117) and a table (above)

Point 5:

Discussion

- Among the study’s Limitations the Authors should mention the non-randomized nature of their comparisons, and – more generally - the issue of comparability of the patients’ groups, particularly for unmeasured variables. Moreover the Authors should not interpret the results which are statistically significant differences as being clinically important.  

 Response 5: 

It is true that patients were divided into groups according to the presence or absence of cognitive impairment and perhaps it cannot be considered randomized, but this was determined by the purpose of the study. However, we did add a limitation section (line 199).  The groups were comparable with each other, we added a table and now you can see it clearly. We agree that it is incorrect to define the statistically significant results obtained as clinically important and have removed these sentences so as not to impose this opinion.

Point 6:

Other comments

- At page 3, line 90  ‘abnormal distribution’ should be ‘non-normal distribution’.

Response 6: 

Thank you, we have corrected the technical error

Summary comments

This study may be of potential interest: however the current version has significant issues to be addressed, needing a profound revision.

Thank you for your interest, comments, and attention to our work.

Reviewer 2 Report

Minor comments for authors:

Line 40 use "extent" instead of extend.

Line 44 call out MRS  magnetic resonance spectroscopy (MRS) 

Line 71 need to call out the abbreviations of examined variables i.e.

N-acetylaspartate (NAA), choline (Cho), creatine (Cr) (Cr2), and phosphocreatine (Cr2)

Author Response

Point 1: Line 40 use "extent" instead of extend.

Response 1:  Thanks for the correction

Point 2: Line 44 call out MRS magnetic resonance spectroscopy (MRS) 

Response 2: corrected

Point 2 : Line 71 need to call out the abbreviations of examined variables i.e. N-acetylaspartate (NAA), choline (Cho), creatine (Cr) (Cr2), and phosphocreatine (Cr2)

Response 3: corrected

Thank you for your interest, comments, and attention to our work.

Round 2

Reviewer 1 Report

I thank the Authors for addressing the comments and revising the manuscript which is now improved. I  have few residual comments.

1. I would recommend to justify the chosen sample size. The explanation provided by the Authors is still unclear: even when data follow a non-Normal distribution sample size can be determined. The above justification is particularly important for data interpretation of study findings which cannot reach statistical significance.

  1. The fact that no cases of missing data were present is an important, positive feature, which should be specified in the Statistical Analysis Section.

  1. I would also suggest to use multivariable models: this option would improve interpretability of study results.

Author Response

Thank you for your comments! We have added the necessary information to the statistics section. We will be sure to take all requirements into account in our future research